# Efficient Toughening of Short-Fiber Composites Using Weak Magnetic Fields

**DOI:** 10.3390/ma13102415

**Published:** 2020-05-25

**Authors:** Omri Goldberg, Israel Greenfeld, Hanoch Daniel Wagner

**Affiliations:** Department of Materials and Interfaces, Weizmann Institute of Science, Rehovot 76100, Israel; green_is@netvision.net.il

**Keywords:** short-fiber composites, fracture toughness, magnetic orientation, magnetic translation

## Abstract

Short fibers may serve as toughening agents of composite materials because of the high energy dissipated during fracture, associated with numerous fiber pullouts. An ongoing challenge is to improve their toughness even further, by directing and concentrating fibers near highly stressed structural regions. Weak magnetic fields are utilized to increase the fracture toughness of an epoxy matrix reinforced by short magnetized glass fibers by directing and concentrating fibers near highly stressed structural regions. The orientation and local concentration of the fibers are controlled by the vector components of the magnetic field, and by the gradient in field intensity, respectively. Optimized fracture toughness was achieved by using two pairs of permanent magnets, combining enhanced concentration of fibers in the crack-tip vicinity with alignment of the fibers along the load direction. This optimized value was well above the reference fracture-toughness measured for composites with the same filler content in the absence of a magnetic field, as well as above the value achieved by exploiting unidirectional alignment, without fiber translation, using a solenoid. The method suggested in this study—localized reinforcement using magnetic translation of fillers through the formation of magnetic gradients—enables efficient and controllable improvement in the composite’s overall resistance to fracture, without the involvement of additional phases or material.

## 1. Introduction

The mechanical properties of fibrous composites are controlled by a few parameters that can be roughly classified into two categories: (1) the properties of the matrix and the fibers individually, including: moduli, strengths, fiber length and orientation distribution, aspect ratio and volume fraction; and (2) the performance of the fiber–matrix interface, for example: the interfacial bonding strength, voids and friction. Controlling these two aspects paves the way for defining a representative volume element [1] (RVE) which outlines the collective properties of the structure as a whole. Then, according to the composite function, it is possible to design it to withstand specific loading conditions, given the average properties of the RVE. However, in practice, the assumption of an RVE is not always sufficient to optimize a composite system; localized stress concentration, due to edges or small imperfections, may result in the propagation of a crack and eventually lead to a catastrophic failure. A historical example of this outcome is the sinking of numerous American “Liberty Ships” during World War II. High stress concentration near the boundary of square edge cargo hatches resulted in the sudden destruction of entire vessels, some of them breaking into two halves [2]. This prominent event highlighted the importance of Griffith’s seminal work [3] (1920) regarding strain energy release rate and its significance to mechanical engineering, promoting the idea that fracture mechanics is governed by the local stress field near the crack-tip. To this day, engineers are facing challenges restraining local stress intensities [4,5]. The solutions often involve additional phases, which introduce new interfaces, essentially shifting the threat of failure to a different location. The control over local stress using translation of reinforcing agents, such as fibers in a continuous phase, can serve as an elegant solution, that optimizes toughness while avoiding the involvement of additional fillers or phases.

Considering linear elastic materials, toughness ($G_{c}$) measures the strain energy released by creating two new surfaces when a crack extends. However, as described by Ritchie [6], two classes of mechanisms can dramatically increase toughness: (i) Intrinsic toughening mechanisms, associated with damage-inhibiting processes within a limited volume around the crack known as the process-zone. These mechanisms are generally associated with plasticity and are effective in preventing crack initiation; (ii) The presence of fibers near the crack-tip generates additional contributions of energy dissipation during fracture, known as extrinsic toughening mechanisms [6]. These include processes such as fiber pull-out, fiber bridging, fiber debonding and stress redistribution [7]. These extrinsic toughening mechanisms are activated ahead of the crack-tip, therefore they do not take part in crack initiation. Toughening by fiber alignment relies on the fact that these mechanisms are sensitive to the angle between the main axis of the fiber and the crack trajectory [8,9,10]. In particular, short fibers serve as excellent candidates for composite toughening as they offer rapid and low-cost production [11]. Additionally, in contrast to continuous fibers that tend to rupture during composite fracture, short fibers have higher probability for pullout events that are accompanied by high-energy dissipation mechanisms [12].

An alternative measure of strain energy release rate (G, commonly measured in kJ/m^2^) is the stress intensity (K, measured in Pa·m^0.5^). Unlike energy, which is a scalar, *K* is separable into its three spatial components (or modes). For example, in Mode I of a fracture, also known as the opening mode, the load acts on the vector normal to the crack wall’s surface and is perpendicular to the crack propagation trajectory. Another trait of stress intensity is its association with the spatial distribution of stress in the crack near-field. Accordingly, the stress distribution in the vicinity of a crack-tip is expressed in polar coordinates (r,θ) for its first order, as follows:(1)$$\sigma_{ij}\left(r,\theta \right)=\frac{K}{\sqrt{2\pi r}}f_{ij}\left(\theta \right)$$
where $f_{ij}$ is a dimensionless function of the angle θ which varies with load and geometry. The fracture toughness ($K_{c}$) denotes the criterion for crack propagation when the stress intensity K at the crack-tip overcomes a critical value (similarly, the toughness $G_{c}$ stands for the critical value of the energy release rate G). Similar to toughness ($G_{c}$), fracture toughness is considered a material property, and in some cases the two quantities are interchangeable. Assuming self-similar crack growth in opening mode (also termed Mode I) under plane strain, that relation can be written as [13]:(2)$$G_{c}=\left(1-\upsilon^{2}\right)\frac{K_{Ic}{}^{2}}{E}$$
where E is the elastic modulus and υ is the Poisson ratio.

In previous work, we demonstrated that spatial control of magnetized glass fibers (MGFs) using weak magnetic fields improves the elastic modulus of composite cylinders under bending [14]. In general, the stiffness of a composite is determined by the weighted sum of its components, which limits the application of filler density control to specific cases only. In contrast, when an object deforms beyond its elastic regime and a fracture develops, the mechanics of crack initiation and propagation are no longer described by averaged values, and the system is shifted to a “weakest link” regime, strongly dependent on the crack-tip near-field stress. Therefore, the application of magnetic fields to locally concentrate and direct fibers seems highly suitable to efficiently improve toughness.

The magnetization ($\vec{M}$) of a material in response to an applied field ($\vec{H}$) is defined by the magnetic susceptibility ($\vec{M}=\chi \vec{H}$). The magnetic susceptibility is the slope of the linear part of the M–H curve at relatively low fields; materials that possess susceptibility of χ<0 are classified as diamagnetic materials—the absolute value of their susceptibility is extremely low and accordingly they do not display any significant response under common magnetic sources. Materials with susceptibility of χ>0 tend to show magnetic responses under relatively low magnetic fields, and they can be classified into categories (such as ferromagnets, ferrimagnets or paramagnets) according to the degree of order of their magnetic domains. Ferromagnetic and ferrimagnetic materials are characterized by a hysteresis relationship between their magnetization and the applied field, and as a result they retain some magnetization after removal of the magnetic source. Paramagnetic materials, on the other hand, do not possess a remnant magnetization. As most common fillers are diamagnetic and carry low and negative susceptibility, they respond to extremely strong magnetic fields, which are not viable for large-scale applications. Magnetite (iron-oxide) is typically considered a ferrimagnet, in the form of a nanoparticle, however, the effect of the magnetic domain order is cancelled, and it transforms into a superparamagnet, which is characterized by the absence of remnant magnetization. Utilization of weak magnetic fields (B < 1 Tesla) requires some magnetization of the fillers. Coating of fibers with small amounts of superparamagnetic iron-oxide nanoparticles (SPIONs) proved to be a convenient method to magnetize diamagnetic fillers while preserving their mechanical attributes [15,16]. Application of magnetic fields during the curing process allows rotation and translation of magnetized fillers in a fluidic medium; after the matrix solidifies, the fillers remain frozen in their final state. Unidirectional orientation of fibers in a composite can increase the composite elastic modulus [17,18] and strength [18], as well as its toughness [19,20,21]. Whereas rotation of magnetic fibers is dictated by the vector components of the magnetic field, magnetic translation is actuated whenever there is a local gradient in the total field intensity [22].

Controlling the orientation of fillers by magnetic fields has been demonstrated in the past [23,24]; over the years, weak magnetic fields have been utilized [25,26], and the method was further developed by Studart to create non-uniform structures of filler orientation [27,28,29,30]. However, these previous works did not demonstrate translation of fibers for the specific purpose of locally varying filler concentration. In this work, we introduce magnetic translation, using field intensity gradient as a supplementary tool, to design non-uniform structures as a means to toughen a standard composite of epoxy and short MGFs. A magnetic field was applied during the curing process to shape the structure of the composite in terms of fiber local density and orientation. To analyze both aspects, different schemes of magnetic fields were investigated as an attempt to manipulate, individually and collectively, the vector components of the magnetic field and the spatial gradient of its intensity. Fiber concentration near the crack-tip was visualized optically, owing to the strong light extinction of magnetite nanoparticles adsorbed on the magnetized fibers. Fiber orientation angle and length were measured by Computed Tomography. Scanning Electron Microscopy was used to assess the different toughening mechanisms involved during fracture.

## 2. Experimental

### 2.1. Magnetization of Glass Fibers

Milled glass fibers (GF) (Microglass-3032, Fibertec, diameter = 16 μm, average length = 220 μm) were chosen to serve as toughening agents in a short-fiber-reinforced polymer (SFRP). The fibers were coated with magnetite nanoparticles (d = 10 nm, χ = 3.02) by physical adsorption of a cationic ferrofluid (EMG-605, FerroTech Santa Clara, CA, USA) (Figure 1). The ferrofluid was added in a ratio of 70 μL per 1 g of fibers to a vial containing 20 mL of deionized water. To estimate the effect of fiber interface on the adhesion of the magnetite particles with the Epoxy matrix (EP), an additional set of Silane-sized magnetized short glass fibers was tested as well (Microglass-9132, Fibertec, Bridgewater, MA, USA). The mixture was slowly rotated at 3 RPM for 48 h. The fibers, which by then changed their color from white to brown, were filtered and washed three times with deionized water. After final drying of the powder at 110 °C for 5 h, a small sample was taken to a Superconducting Quantum Interference Device (SQUID) (MPMS-3, LOT-Quantum Design, Darmstadt, Germany) to measure the MGF magnetic susceptibility (χ) and magnetization saturation (M_s_).

### 2.2. Composite Preparation

A diglycidyl ether bisphenol-A (DGEBA)-based epoxy matrix (EP-502, Polymer G) was selected as the matrix of the SFRP due to its low viscosity, long curing time and low strain-to-failure [21]. Short glass fibers were mixed in a vial containing the epoxy resin and degassed under vacuum (4 h) to eliminate air bubbles. Crosslinking of the matrix was carried out with a compatible curing agent based on diethylene tetramine (DETA) (EPC-9, Polymer G) in a 16:100 weight ratio. After additional degassing of 30 min, the mixture was cast in a compact-tension (C-T) shaped silicon mold. Next, the mold was covered with a thin silicon sheet and sealed inside a small aluminum container. Curing was performed at room temperature overnight while the specimens were positioned in pre-determined magnetic conditions according to the specimen type, as further elaborated in Section 2.3. After overnight curing all specimens were post-cured inside a furnace for four hours at 80 °C.

### 2.3. Curing Protocol and Setup

Several setup environments were selected to define different sample categories, as specified in Table 1 and Figure 2. While the categories were differentiated according to the magnetic configuration applied during curing, all the samples consisted of 10 wt % unsized GF/EP composites. As a convention, the three vectors of the magnetic field are defined with respect to the compact-tension sample geometry (Figure 3), where the load direction during testing is *X*, the crack propagates (at opening mode) in direction *Y* and the crack thickness is along the *Z*-axis.

To concentrate the MGFs near the crack-tip, a pair of small permanent magnets were fixed above and below the perimeter of the root crack-tip that is patterned in the mold. This arrangement (setup 1) forces the field lines to align in the *Z* direction between the two magnets (Figure 2), and increases the field intensity |H| ($\sqrt{{\vec{H_{x}}}^{2}+{\vec{H_{y}}}^{2}+{\vec{H_{z}}}^{2}}$) locally around the crack-tip, thus generating a gradient in field intensity which is essential for fiber translation.

In a second curing setup (setup 2), aimed to assess the effect of unidirectional orientation of the fibers in the *X*-axis, the sample was rotated around its *X*-axis inside the air-core of a homemade solenoid (2 mT, 4 Ampers). As no magnetic force was acting along the *Z*-axis, rotation of the sample was required to avoid gravitational precipitation of the fibers.

To combine the two effects of magnetic concentration and magnetic orientation, the first setup was modified (setup 3). A pair of large Neodymium magnets, applying a magnetic field along the *X*-axis of the specimen, was placed perpendicular to the concentration magnets (Figure 2). The Neodymium magnets served as a bias field (*H_b_*), opposing the field along the *Z*-axis applied by the concentration magnets (*H_c_*), and driving the fibers to rotate towards aligning with the *X*-axis. The final orientation angle θ between the main axis of the fibers and the *X*-axis was predicted to be [31]:(3)$$\theta =tan^{-1}\left[{\left(\frac{H_{c}}{H_{b}}\right)}^{2}\right]$$

Control over the distance between the two concentration magnets, and thus over the intensity of *H_c_*, enabled the tuning of θ and ∇|H| near the crack-tip. As the ratio *H_c_*/*H_b_* increases, fiber localization is enhanced at the expense of fiber orientation; when the ratio is greater than zero the angle θ deviates from perfect alignment with the *X*-axis. To optimize the net result, various *H_c_/H_b_* ratios were tested in an attempt to maintain a uniform bias field along the specimen while achieving a profile of ∇|H| which is as similar as possible to setup 1 (Figure 4). For comparison, an additional set of samples was cured while rotating in the absence of a magnetic field. The magnetic field distribution in all configurations was measured using a Gaussmeter (Lake-shore 460). Photographs of the curing configurations are shown in the Appendix A.

### 2.4. Notching

The root notch of the post-cured compact-tension sample was sharpened using a V-shaped disc, forming a machined notch with a radius of curvature of ρ ≈ 50 μm (Figure 3). Finally, a natural notch was generated using a homemade guillotine (Figure 3); a weight stroke a fresh blade positioned inside the machined notch, leading to a natural crack with a radius of curvature of a few microns (Figure 3).

### 2.5. Mechanical Testing

The fracture toughness ($K_{Ic}$) of GF/EP composites was measured using standard compact-tension specimens [32] (refer to Figure 3: W = 20 mm, a = 8 mm, b = 7 mm), tested with an Instron-5965 instrument using a 5 kN load-cell at a rate of 1 mm/min. The load and displacement were recorded by a compatible software (Bluehill 3) and the fracture toughness was computed as follows:(4)$$K_{Q}=\left(\frac{P_{Q}}{bW^{1/2}}\right)f\left(x\right),\;f\left(x\right)=\frac{\left(2+x\right)\left(0.886+4.64x-13.32x^{2}+14.72x^{3}-5.6x^{4}\right)}{{\left(1-x\right)}^{3/2}},$$
where $P_{Q}$ is the maximum load as required in ASTM D5045-99 Section 9.1.1, and x=a/W. The calculation of fracture toughness for specific crack configurations is a central problem in theoretical fracture mechanics; some authors [33] consider the rigorous fracture mechanics approach to be inapplicable to the case of short-fiber composites. Indeed, difficulties arise because infinitesimal crack growth is a prerequisite in the process of testing crack stability in homogenous materials. In inhomogeneous materials (composites), crack growth may seem stable on average, however, it often comprises sudden cracking events in one of the phases. As a result, some of the energy is released in the form of kinetic energy, and the calculated fracture toughness actually represents a lower bound [9]. Despite the fact that the polynomial in Equation (4) is valid for an elastic-homogenous-isotropic specimen under plane strain, in this study the method was extended to anisotropic specimens, as in other studies [20,21,34], using the direct measurement of *P_Q_* (the load required for crack propagation) as a valid indicator of *K_IC_* [9,33].

The following sets of specimens were tested: (1) composites with randomly oriented fibers for a range of fiber concentrations (Table 2), (2) composites of fibers with different interfaces (Appendix A), (3) composites with fixed filler concentration (10 wt %) which were cured under different magnetic configurations (Table 3), and (4) pure epoxy (no filler) as a reference. Each set of measurements was performed using at least six specimens. To characterize the mechanical properties of the matrix, “dog-bone” samples (length = 15 mm, width = 1.5 mm, thickness = 1 mm) were molded with the EP-502 resin and the EPC-9 curing agent in a ratio of 100:16. The elastic modulus, yield stress (as the intercept of the curve with a 0.2% offset line) and ultimate strength were measured using Instron-5965 (1 mm/min).

### 2.6. Imaging

The distribution of fibers inside the matrix was extracted by microXCT-400 (XRadia, Pleasanton, CA, USA) (working conditions: V = 40 kV and W = 8 W). Raw data was reconstructed with the XRadia software; ortho-slice and three-dimensional visualization were carried out with Avizo software (VSG). Information regarding the distributions of the fiber orientation was retrieved using the cylinder correlation module followed by the “trace correlation lines” tool. Statistics were collected from a representative volume of 2 mm^3^ per sample. The dimensions and morphology of the fibers were evaluated by scanning electron microscopy (Supra55, Zeiss, Oberkochen, Germany). Short glass fibers (in powder form) were scattered over a carbon-tape and coated with thin layer of Pd/Au (S150 sputter-coater, Edwards). Images were taken with a working distance of 6 mm and electron energy of 3 keV. Post-testing fractographic images were taken under the same conditions. Information regarding fiber concentration was retrieved by photographing the compact-tension specimens on a light-table, using a high-quality camera. As specimen photography relied on the light extinction of the magnetite nanoparticles, it was used as a qualitative tool to verify the scale of the concentrated zone. Light extinction is not solely a function of the magnetite concentration but is also sensitive to the particles shape and size. Considering the different orientation configurations, which result in different extinction events, and the degradation of the ferrofluid batch, the concentration distribution of the filler was evaluated only within each sample, and not as a comparison between different samples.

### 2.7. Micromechanics

The mechanical properties of the glass fibers were measured by performing tensile tests of individual fibers (gauge length = 30 mm), using Instron-5965 with a 10 N load-cell at an elongation rate of 1 mm/min. The tensile modulus was calculated as the slope of the stress-strain curve, and the strength was analyzed using a probabilistic two-parameter Weibull distribution of the maximum stresses. The GF’s diameter was ascertained by Scanning Electron Microscopy (SEM), and its length distribution by Computed Tomography. The interfacial shear strength ($\tau_{i}$) and the fiber-matrix surface energy ($G_{i}$) were obtained from pull-out tests of bare GFs and MGFs using a homemade setup. Samples were prepared by embedding a single fiber into a pre-cured epoxy droplet located inside a hollow aluminum screw. Subsequently, a clamped single fiber was carefully inserted into the droplet. The pull-out test was performed using a stiff frame at a cross-head speed of 1 µm/s, while the force-displacement was monitored via computer interface. Both properties were calculated as follows:(5)$$\tau_{i}=\frac{F_{max}}{A_{s}}=\frac{F_{max}}{\pi d_{f}l_{emb}}\;,$$
(6)$$G_{i}=\frac{AUC}{A_{s}}=\frac{1}{\pi d_{f}l_{emb}}\int_{0}^{x_{max}}Fdx\;,$$
where $F_{max}$ is the maximal applied load acting on the fiber, *A_s_* is the fiber surface embedded area, $d_{f}$ is the fiber diameter, $l_{emb}$ is the fiber’s embedded length and AUC is the area under the curve of the force-displacement plot.

## 3. Results and Discussion

### 3.1. System Characterization

MGFs with and without sizing were prepared with various ferrofluid/GF ratios ranging from 40 μL/g to 150 μL/g, for magnetic characterization using SQUID. M–H curves for all samples depicted a superparamagnetic behavior, without any remnant magnetization. Both magnetic saturation (*M_s_*) and susceptibility (*χ*) increased with the ferrofluid content in a monotonic increasing fashion, with no clear difference between the sized and unsized fibers (Figure 1a). A ratio of 70 μL ferrofluid per 1 g of milled GF (*M_s_* ~0.5 emu/g, *χ* ~3 × 10^−3^ cm^3^/g) was selected for further experiments (Figure 1c). SPION adsorption on the fiber surface was visualized using SEM as a dispersed thin layer of nanoparticles; samples of high ferrofluid/GF ratio, on the other hand, exhibited bulky thick layers that covered the glass surface entirely (Figure 1b).

To exclude the effects of fiber sizing and magnetic coating on mechanical properties, four sets of fillers inside the EP matrix were compared: (a) bare GF, (b) unsized MGF, (c) sized GF and (d) sized MGF, as elaborated in Section 3.4.

### 3.2. Magnetic Translation and Rotation

Filler concentration in the crack-tip vicinity was increased by producing a non-uniform field, which according to Ampere’s model exerts a magnetic force $F_{m}$ [22]:(7)$$F_{m}=\mu_{0}V\left|M\right|\nabla \left(\left|H\right|cos\varphi \right),$$
where $\mu_{0}$ is the permeability constant, V is the MGF volume, |M| is the magnetization of a MGF, |H| is the total magnetic field intensity and φ is the angle between the *M* and *H* vectors. Equation (7) shows that the magnetic force is a function of the angle φ, and is therefore followed by alignment of the magnetic dipole with the field direction. The magnetic torque $\tau_{m}$ acting on a magnetic dipole to minimize the angle φ is as follows:(8)$$\tau_{m}=\mu_{0}V\left|M\right|\left|H\right|sin\varphi .$$

In contrast to the magnetic force, the torque is independent of spatial variation in the magnetic field (∇|H|). As a result, by exerting a uniform field at an angle of φ>0 between the field and the fiber main axis (which is the case in setup 2), it is possible to rotate the magnetic dipole without applying any translation force to the entire body. To predict the final structure of the composites, a profile of the field intensity |H| and its vector components ${\vec{H}}_{x}$ and ${\vec{H}}_{z}$ were calculated using the magnetostatics module of the COMSOL Multiphysics software (Figure 4). The calculation was then confirmed for each of the curing configurations, by measurement of the magnetic flux density components along the *X*-axis of the specimen using a Gauss meter (Figure 2). To ensure the existence of a gradient in field intensity, it is essential to keep in mind that |H| is the sum of all vector components (Figure 4d) and that the strong bias field *H_b_* is not uniform along the *X*-axis (Figure 4c). The magnetic flux density between two permanent magnets is non-uniform and reaches a minimum value right between them; the minimal value between the two bias magnets in setup 3 will be denoted as *ΔH_b_* (Figure 4c). As a result, setup 3 generates two opposing gradients of the field intensity at the center of the specimen: a positive gradient, created by H_c_, acting to concentrate the fibers at the center, and a negative gradient, created by *ΔH_b_*, acting to translate the fibers away from the center of the specimen. Therefore, the condition *H_c_* > *ΔH_b_* must be satisfied for fiber concentration.

High concentration of MGF near the crack-tip of the relevant samples was visible to the naked eye as a dark brown circle. The high optical extinction of the SPION, which gave the MGFs their brown hue, served as a convenient indication of local fiber concentration, as clearly seen in the photograph and contour map in Figure 5a,b respectively. Although the data retrieved by photography is mostly qualitative and suffers from a lens-flare around the notch, a greyscale profile curve taken across the *X*-axis suggests that the radius of the concentrated zone varies between 2 and 4 mm. The size of the concentrated zone, indicated on the photograph, is essential to the validity of the compact-tension test, as a large concentrated area implies that the singularity-dominated zone has effectively uniform volume fraction [13,35]. The $K_{IC}$ calculation [Equation (4)] assumes plane strain conditions near the crack-tip [34], which require the concentrated zone to be larger than the radius of plasticity, where the matrix reaches its yield point (*r_p_* ≈ 30 μm). Another meaningful comparison is to the process-zone of toughening mechanisms, estimated for the current composite to be approximately 650 μm [34,36].

Fiber analysis using Computed Tomography confirmed alignment of the fibers parallel to the *Z*-axis for composites cured in setup 1 (concentration magnets), and alignment of the fibers parallel to the *X*-axis for composites cured in setup 2 and 3 (unidirectional orientation using a solenoid and concentration magnets with a bias field, respectively). The average angle between the fibers and the *X*-axis for each configuration was computed according to Equation (3) and given in Table 3. The fiber orientation in the composites is visualized in Figure 6.

### 3.3. Mechanical Analysis 

Fracture toughness ($K_{IC})$ represents the balance between the stored elastic energy, that is released during crack propagation, and the resistance to fracture, generally provided by the surface energy of the crack walls.$K_{IC}$ mainly involves two material properties: the elastic modulus E and the toughness $G_{c}$, as manifested in the relation $K_{IC}\propto \;{\left(EG_{c}\right)}^{0.5}$. By modifying the expression for short-fiber composites, $K_{IC}$ may be expressed by material parameters of the matrix and the filler, including the toughness and the elastic moduli of each phase, the fiber volume fraction ($V_{f}$), the pullout energy and the fiber length distribution. The involvement of magnetic fields during the curing process adds an additional set of controllable parameters: the orientation factor $\eta_{\theta }$, and the concentration coefficient $\eta_{c}$.

The orientation factor $\eta_{\theta }$ is a function of the angle θ between the fibers and the X-axis; it varies between 0 (θ = 90°) and 1 (θ = 0°). As fiber alignment causes $G_{c}$ and E enhancement [12,18,25,34,35,36], it directly affects fracture toughness. For the elastic modulus E, the orientation factor decreases with orientation angle as described by the Krenchel factor—$\cos^{4}\left(\theta \right)$ [37]. The effect of orientation on the toughness $G_{c}$ is more complex. On the one hand, a decrease in pullout energy with increasing angle is expected as the number of fibers crossing the plane decreases. On the other hand, pullout energy may increase with θ up to a maximum value at 45 °C, due to a “snubbing effect” which induces energy dissipation when fibers are pulled out at inclined angles [38,39]. In this study, the snubbing coefficient is too low to compensate the decrease of energy with orientation angle, and the orientation factor for toughness is approximately cos(θ) [10] (S5).

The concentration coefficient $\eta_{c}$ represents the effect of non-uniform filler concentration. It is determined empirically and varies between 0 and $1/V_{f}$. To describe the effect of fiber concentration, we used the term effective volume fraction ($V_{f}{}^{eff}$), expressing the apparent volume fraction when the filler is distributed non-uniformly. For example, a composite with a global volume fraction $V_{f}$ (total filler volume/total composite volume) may exhibit a fracture toughness analogous to a denser composite with a uniform volume fraction of $\eta_{c}V_{f}$ ($\eta_{c}>1$). Accordingly, a composite with concentration coefficient $\eta_{c}=1$ has a global volume fraction that is equal to the apparent $V_{f}$, suggesting a uniform filler distribution. Whenever $\eta_{c}$ is greater than 1, the crack will behave as though the volume fraction it experiences is higher than the global volume fraction [estimations of $\eta_{c}$ are illustrated in the Appendix A].

The fracture toughness improvement was quantified by normalizing with respect to the fracture toughness of the matrix ($K_{IC,0}$). The fibers’ contribution to fracture toughness of the composite is given by the following ratio:(9)$$\frac{K_{IC}}{K_{IC,0}}=\sqrt{\frac{G_{c}E}{G_{c,0}E_{0}}}≅F\left(\frac{E_{f}}{E_{m}},\frac{G_{c,f}}{G_{c,m}},\;V_{f},\;\eta_{\theta },\eta_{c},\eta_{l}\right)\;,$$
where E, $G_{c}$ and $K_{IC}$ are the elastic modulus, toughness and fracture toughness of the composite, respectively. The subscript *0* refers to the measured value of the pure matrix (no filler), and the subscripts *m* and *f* refer to the contribution of the matrix and of the fibers, respectively. $\eta_{l}$ is the efficiency coefficient for a given fiber length. The fracture toughness is enhanced when $\frac{E_{f}}{E_{m}},\frac{G_{c,f}}{G_{c,m}}$ or the volume fraction is increasing, and grows monotonically with the efficiency factors $\;\eta_{\theta },\eta_{c}$ and $\eta_{l}$ (see Appendix A). The function F cannot be expressed by a plain rule of mixture of fiber and matrix contributions, as it includes cross coupling terms between the modulus and the toughness.

The viscosity of the resin, which increases over time until the medium solidifies, damps the migration and rotation of the fibers. Nevertheless, the curing time is sufficiently long to bring the current system to equilibrium [31]. As a result, the degree of toughening can be optimized experimentally solely by tuning the magnetic field sources involved in the curing process. By increasing the ratio *H_c_/H_b_* up to a point where *H_c_ >> ΔH_b_*, the concentration efficiency will reach a maximal value, whereas the orientation efficiency will decrease according to Equation (3).

### 3.4. Mechanical Measurements

The effect of the filler–matrix interface was considered first. The interfacial shear strength ($\tau_{i}$) between the matrix and the fibers, measured in pullout tests as the slope of the force versus the interfacial area, showed no significant difference between GFs and MGFs (Figure 7a). As an additional step, four types of fiber (pristine GF, unsized MGF, sized GF and sized MGF) were used as fillers in epoxy composite compact-tension specimens with three distinct weight fractions (2%, 10% and 20%). No significant effects on fracture toughness were observed for all measured concentrations, regardless of the presence or absence of fiber sizing (Figure 7b, Appendix A). Subsequently, the rest of the present work was performed exclusively with unsized fiber.

The relation between weight fraction and $K_{Ic}$ was tested for weight fractions up to 30 wt %, and the composites’ toughening was calculated relative to the fibreless epoxy matrix ($K_{Ic,0}=0.86\pm 0.21$ MPa·m^0.5^). Consistent toughening of the composites was detected as the filler concentration exceeded 10 wt %, confirming the assumption that higher fiber concentration toughens the composite (Figure 8). Table 2 summarizes the measured toughness and the normalized toughening as a function of fiber content. The present work focused on samples with fiber concentration of 10 wt %, showing, as a proof of concept, a significant mechanical improvement using the proposed method even at relatively low fiber content.

To assess the mechanical implications of the structures designed with the assistance of magnetic fields, compact-tension tests were conducted for seven sets of 10 wt % MGF/EP composites, each cured under a different magnetic configuration. All fracture toughness measurements of the various magnetic configurations are summarized in Figure 9 and Table 3.

Compact-tension tests of samples cured in setup 1 (no bias field, *H_c_*/*H_b_*→∞,) exhibited a relatively low fracture toughness with normalized value of $K_{Ic}/K_{Ic,0}$ = 0.93, suggesting negative contribution to toughness compared to the control specimens cured without any magnetic field ($K_{Ic}/K_{Ic,0}$ = 1.12). Whereas random orientation activated a mild toughening mechanism, the alignment along the *Z*-axis imposed on the fibers by the concentrating magnets weakened their contribution to composite toughness. Although the local weight fraction of the fibers near the crack-tip was greater than 10%, suggesting $\eta_{c}$ > 1, as observed visually (Figure 5 and Appendix A), their presence was inconsequential in terms of toughness. This is expected for an orientation of θ = 90°, as it leads to $\eta_{\theta }$ ≈ 0 and, as a result, to no fiber contribution to toughness [10] ($\eta_{\theta }\eta_{c}$ ≈ 0). In fact, the value measured was slightly lower than $K_{Ic}$ measured for pure epoxy, implying a possible weakening effect, probably due to high stress concentrations near fiber ends.

Composites cured in setup 2 (a solenoid, *H_c_/H_b_*
*= 0*) exhibited considerable toughening ($K_{Ic}/K_{Ic,0}$ = 1.56), significantly higher than the control ($K_{Ic}/K_{Ic,0}$ = 1.12). Notably, the extremely low magnetic field induced by the copper coil (*H* = 30 *G*) was sufficient to align the short fibers, as verified by micro-Computed Tomography (μCT) imaging (Figure 6b), which serves as evidence of the high susceptibility of SPION and the low energy consumption required for producing the magnetic field. As expected, alignment towards the *X*-axis positioned the main axis of the fiber perpendicular to the crack trajectory and ensured maximal exploitation of energy dissipation ($\eta_{\theta }$ ≈ 1). As the magnetic field intensity inside the solenoid is approximately uniform, the fiber distribution was uniform as well ($\eta_{c}$ ≈ 1), as visualized by the composite profile in the μCT images.

After establishing that both high fiber concentration and low orientation angle θ intensify fracture toughness separately, our next target was to validate their joint contribution experimentally. The third setup yielded concentrated and aligned samples, using both concentrating magnets and a bias field. Large magnetic ratios of *H_c_/H_b_* > 1 yielded composites which were not as tough as the perfectly aligned samples cured in the solenoid (*H_c_/H_b_* = 0), probably due to the poor orientation efficiency (θ > 45°) caused by the concentration field. On the other hand, weak fields acting along the *Z*-axis (*H_c_/H_b_* = 0.1) did not improve the fracture toughness significantly ($K_{Ic}/K_{Ic,0}$ = 1.60) in comparison to perfect alignment of uniform concentration ($K_{Ic}/K_{Ic,0}$ = 1.56). Any gradient in field intensity should have resulted in migration of fibers towards the crack-tip periphery to some extent; however, it appeared that in practice, the local variation in the bias field intensity (*ΔH_b_*) set a lower bound for the required intensity of *H_c_*. Maximum fracture toughness increase was measured for a ratio of *H_c_/H_b_* = 0.3 with substantial toughening ($K_{Ic}/K_{Ic,0}$ = 1.86). This ratio appears to combine the contributions of both low orientation angle (θ ≈ 5°) and larger filler concentration near the crack-tip.

### 3.5. Toughening Mechanisms

As described in Section 3.3, the improvement in fracture toughness is likely related to an increase in the elastic modulus (according to the rule of mixtures) and to an enhancement of $G_{c}$, which mainly results from fibrous toughening. Focusing on the toughness, several mechanisms are involved, all essentially involving an increase in $\eta_{\theta }$ and $\eta_{c}$ [19,40]. In fibrous composites, the occurrence of the different toughening mechanisms is generally determined by the probability of occurrence of two main failure events: fiber pullout and fiber-rupture. The probabilities for both events are embodied in the critical length of the fiber ($l_{c}$) [12]: a fiber of length $l<l_{c}$ located perpendicular to the crack path will most likely be pulled out from the matrix. For composites with fiber lengths of $l>l_{c}$, the probability for fiber pullout is $l_{c}/l$, so fiber-rupture will become more frequent as the fibers are longer. The critical length is calculated according to [41]:(10)$$l_{c}=\frac{\sigma_{f}{}^{max}d_{f}}{2\tau_{i}}$$
where $d_{f}$ is the fiber diameter, $\sigma_{f}{}^{max}$ is the fiber strength and $\tau_{i}$ is the interfacial shear strength between the fiber and the matrix. As the fibers used in this work are discontinuous and their lengths (~220 μm) are well below the critical length ($l_{c}$ ≈ 740 μm), it was anticipated that during fracture the probability of fiber-rupture would be much lower than the probabilities of debonding and pull-out. Pullout tests of single fibers from the matrix were performed to estimate the interfacial shear strength ($\tau_{i}$) and fiber-matrix surface energy ($G_{i}$), as detailed in Table 4. The force-displacement curve (Figure 10) was characterized by a linear increase in force followed by a sudden drop, suggesting an elastic stress transfer followed by a minor plastic deformation and, finally, failure by fiber debonding and pullout [42]. In a few samples, after debonding, some energy was dissipated as interfacial friction as well. Post-fracture electron microscopy images (Figure 10) of the oriented composite clearly show multiple pulled-out fibers perpendicular to the fracture walls, as well as a few bridging fibers held by the two fracture walls on both ends. Snubbing can also be observed in the bottom image, where fibers oblique to the fracture surface do not break, but instead pull out, while applying lateral pressure on the matrix (see Appendix A).

## 4. Summary and Conclusions

Weak magnetic fields were utilized to tune and maximize the fracture toughness of short-fiber-reinforced epoxy. Toughening via unidirectional magnetic alignment was achieved by curing the resin, under rotation, inside the air-core of a solenoid. Compact-tension samples of 10 wt % MGF/EP showed a significant increase of 56% in *K_IC_* with respect to pure matrix, whereas randomly aligned samples exhibited a smaller increase of just 12%. Magnetic translation was then employed to position the fibers near the crack-tip by generating a gradient in magnetic field intensity during the curing process. Although simple concentration of fibers using permanent magnets was sufficient to increase the effective volume fraction around the crack-tip, it also rotated the fibers so that their contribution to toughness was largely diminished. Integration of both orientation and concentration required the addition of a bias field. By tuning the intensity of both magnetic fields, the concentrating field and the bias field, it was possible to position the fibers near the crack-tip while keeping their orientation angle close to 0°, leading to a toughening increase of 86% with respect to uniformly dispersed randomly oriented fibers.

Magnetism was applied to add two degrees of freedom to the composite fracture toughness: effective volume fraction and orientation. Mechanical properties were tuned by spatial control of the magnetic field vector components and intensity gradient. This new approach enables reinforcement and toughening of materials at specific structural loci, which are more prone to failure, without altering the composition of the entire structure or adding more phases. Composites design by magnetic fields has the potential to enable intricate design of properties on the microscale that may be further applied to unique structures, such as composites with spatially varying stiffness [43], with arrays of defects [44,45] and graded interfaces [46].

## Figures and Tables

**Figure 1 materials-13-02415-f001:**
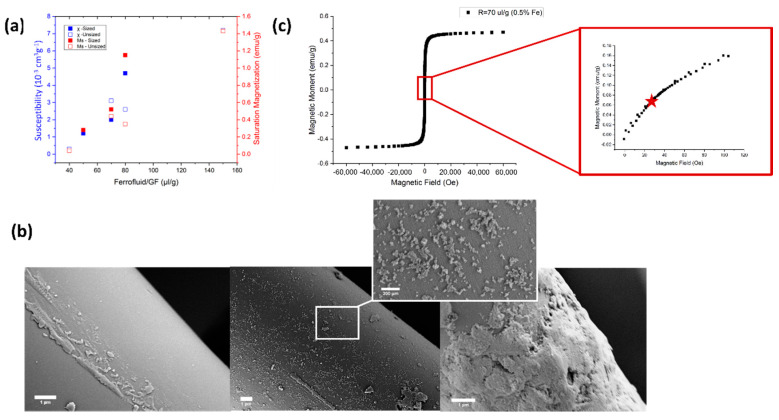
Magnetization of glass fibers. (**a**) The susceptibility (blue squares) and saturation magnetization (red squares) of magnetized glass fibers (MGF) measured by Superconducting Quantum Interference Device (SQUID) as a function of the ferrofluid concentration. Filled and hollow squares symbolize sized and unsized glass fibers, respectively. No effect of fiber interface on nanoparticles adsorption can be seen. (**b**) Electron microscopy images of fiber surface in the absence of magnetite coating (left), after mixing with 70 μL/g of ferrofluid (middle), and after mixing with 150 μL/g (right). In caption: zoomed-in image (scale bar = 300 nm) of the superparamagnetic iron oxide nanoparticles (SPIONs). (**c**) M–H curve for MGF treated with 70 μL/g ferrofluid, corresponding to Fe content of 0.5%, showing paramagnetic behavior. In caption: zoom-in into the domain of low-fields, where the red star indicates the low magnetic field induced by the solenoid.

**Figure 2 materials-13-02415-f002:**
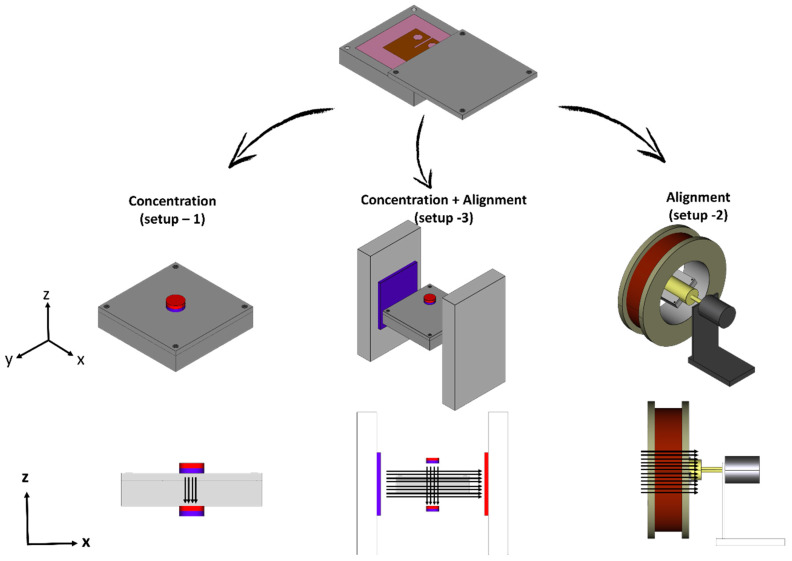
Curing setups of the different magnetic configurations. Top: Pre-cured resin inside a silicon mold sealed by an aluminum container is placed in one of the three setups; another set of samples is cured in the absence of magnetic field as a reference. Middle row: Three-dimensional sketches of the three setups: a pair of concentrating magnets (setup 1), a solenoid (setup 2) and a system of concentrating magnets combined with a bias field. Bottom row: the *XZ*-plane view of the three configurations with black arrows representing the acting field lines.

**Figure 3 materials-13-02415-f003:**
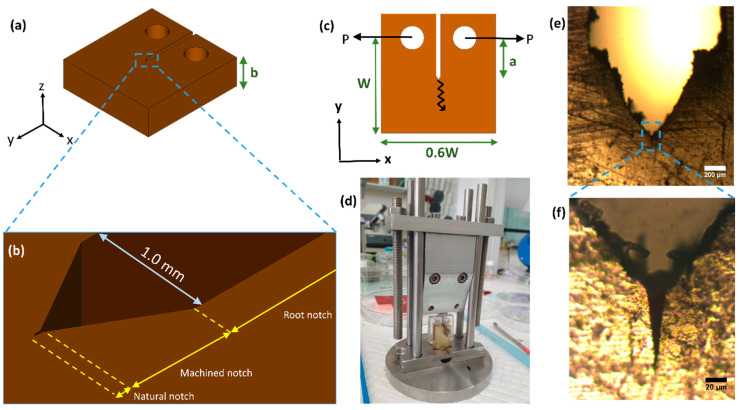
Specifications of compact-tension specimens according to ASTM D5045-99. (**a**) A three-dimensional view where b is the specimen thickness along the *Z*-axis. (**b**) Zoomed-in illustration of the specimen notch. (**c**) A two-dimensional view of the *XY*-plane elucidating the specimen characteristic length W, the initial crack length *a* and the directions of the load (P) and crack propagation (zigzag arrow) during a test. (**d**) A photograph of the Guillotine used to generate the natural notch. (**e**) Light-microscopy image of the V-shaped notch created by a rotating disc. (**f**) Zoomed image of the natural notch created by the Guillotine.

**Figure 4 materials-13-02415-f004:**
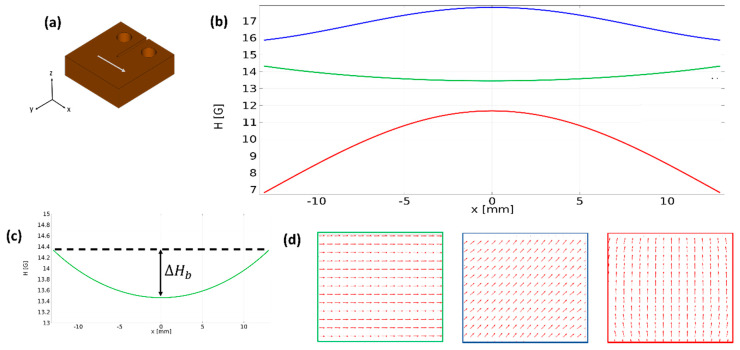
Calculation of the magnetic flux density and vector components using COMSOL MULTIPHYSICS. (**a**) White arrow indicates the directionality of the measurements along the *X*-axis of the specimen. (**b**) The profile of the field intensity in setup 3, involving a bias field and a concentration field. The blue curve indicates the profile of the field intensity, which is a superposition of the concentration field and the bias field. The red and green curves indicate the *Z*-component (originated from the concentration field) and the *X*-component (originated from the bias field) of the field intensity, respectively. (**c**) The profile of the field intensity while using only a bias field; note that the field intensity has some deviation from uniformity, which is maximal in the middle of the two magnets and denoted by Δ*H_b_*. (**d**) The vector field in the XZ plane; the red caption referring to the vector field created by the concentration field, the green caption to the vector field created by the bias field, and the blue caption to the vector field created by setup 3 which is the superposition of the two fields.

**Figure 5 materials-13-02415-f005:**
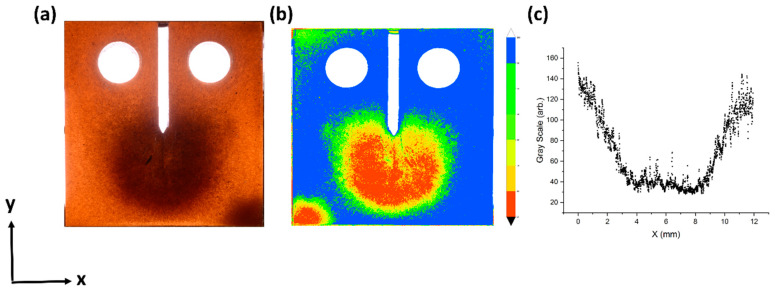
Concentration of fibers in the crack vicinity. (**a**) A photograph of compact-tension specimen cured under the influence of concentrating magnets (setup 1). (**b**) A light intensity contour map of the specimen photograph highlighting the fiber distribution. (**c**) A profile of the photograph greyscale values (0—black, 255—white) along the X-axis taken a few millimeters below the crack-tip.

**Figure 6 materials-13-02415-f006:**
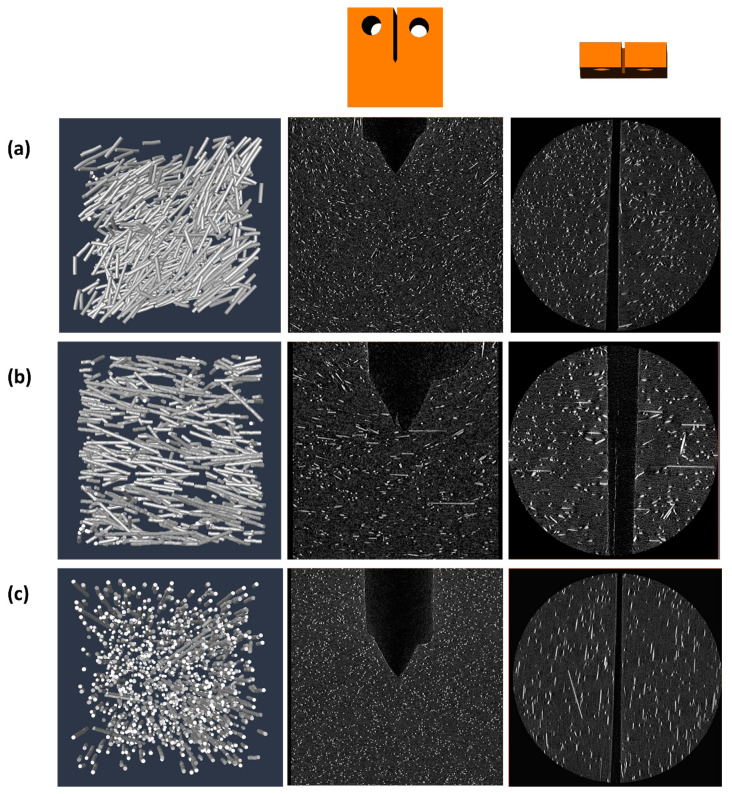
Visualization of fiber orientation by means of Computed Tomography. The left column depicts trace-lines of the fibers from a small representative volume near the crack-tip, the middle column depicts a slice of the *XY*-plane, and the right column a slice of the *XZ*-plane above the crack-tip. Each row displays a different magnetic configuration: (**a**) arbitrary orientation in the absence of magnetic field; (**b**) unidirectional alignment with the *X*-axis, parallel to the direction of the load, by a solenoid; and (**c**) unidirectional alignment with the *Z*-axis, perpendicular to the load and to the crack propagation vector, by concentrated magnets without any bias field.

**Figure 7 materials-13-02415-f007:**
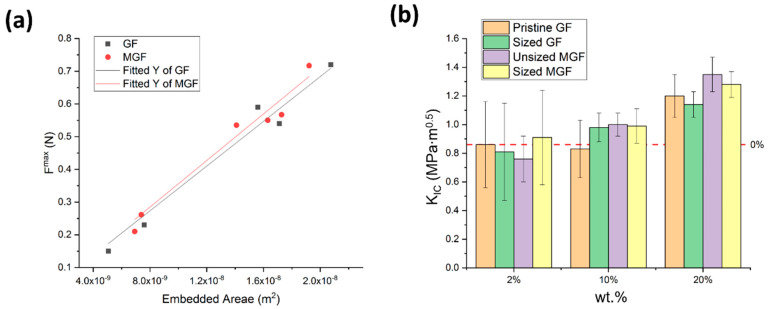
Effect of interface. (**a**) The maximum force required during the test is plotted as a function of the embedded surface area of the fiber, for glass fiber (GF) and magnetized glass fiber (MGF). (**b**) Fracture toughness of epoxy composites reinforced by glass fibers with different interfacial coating and three weight fractions. No magnetic field was applied, and the fibers were uniformly distributed and randomly oriented.

**Figure 8 materials-13-02415-f008:**
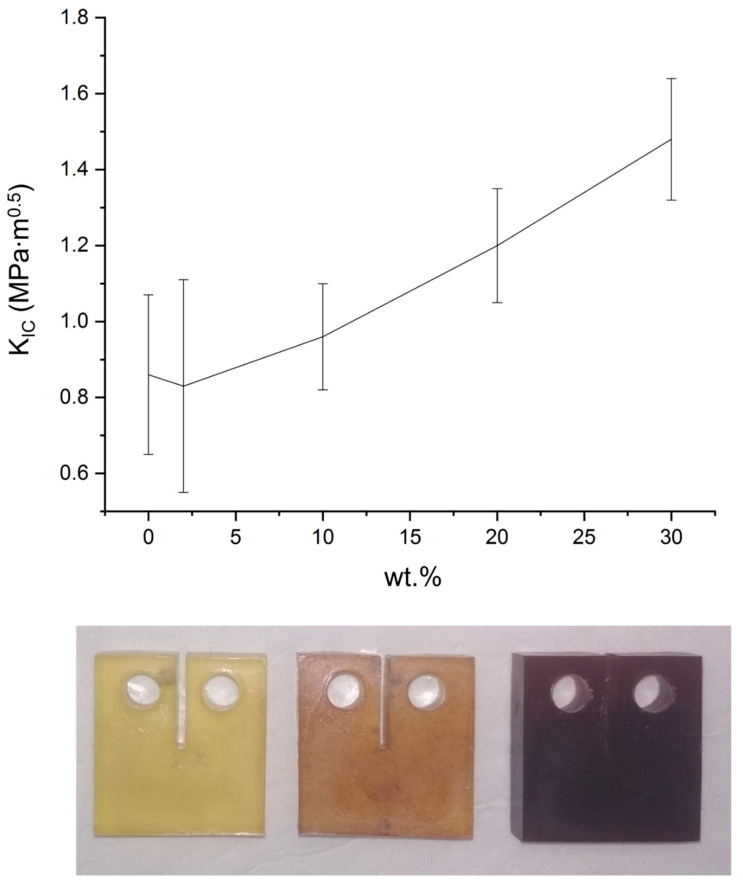
Fracture toughness (*K_IC_*) of epoxy composites reinforced by glass fibers as a function of weight fraction. Bottom: the change in composites color upon addition of MGF for weight fraction of (from left) 0%, 2% and 10%. No magnetic field was applied, and the fibers were uniformly distributed and randomly oriented.

**Figure 9 materials-13-02415-f009:**
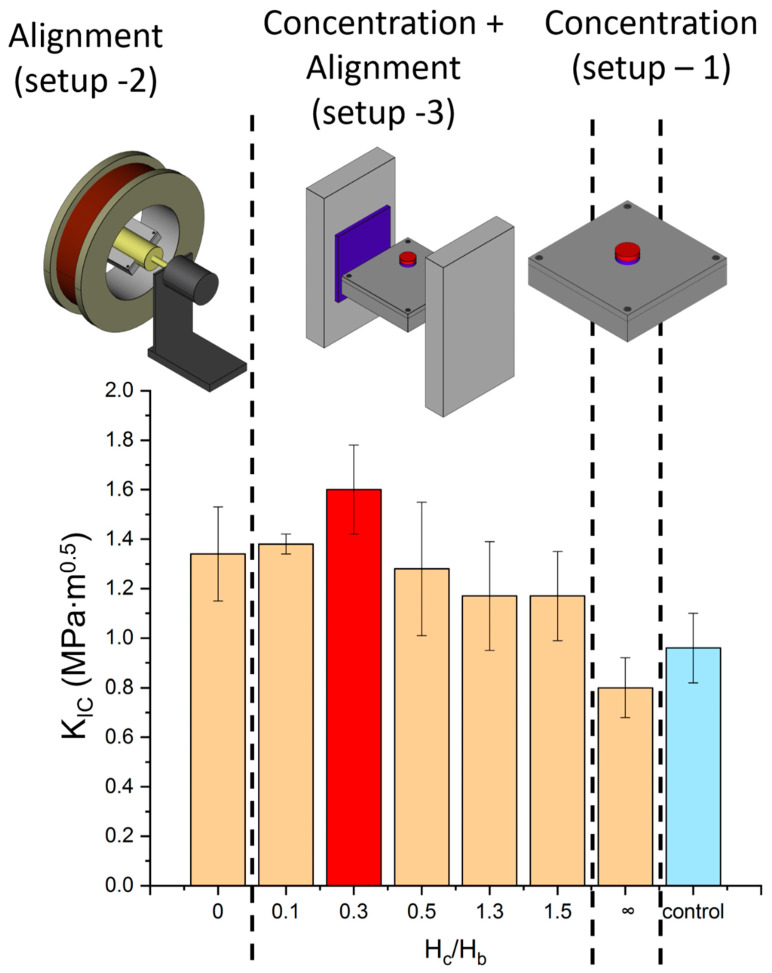
Fracture toughness of 10 wt % MGF/epoxy composites measured after using different magnetic configurations during the curing process. The horizontal axis represents the ratio between the concentrating field acting along the *Z*-axis of the specimen (*H_c_*) and the bias field acting along the *X*-axis (*H_b_*). A ratio of 0 indicates curing without a concentrating field (inside a solenoid as in setup 2), whereas a ratio of infinity indicates curing without any bias field (setup 1). The red column designates the value of *H_c_/H_b_* that maximized fracture toughness. The blue column (control) indicates curing in the absence of magnetic field (uniformly distributed and randomly oriented fibers).

**Figure 10 materials-13-02415-f010:**
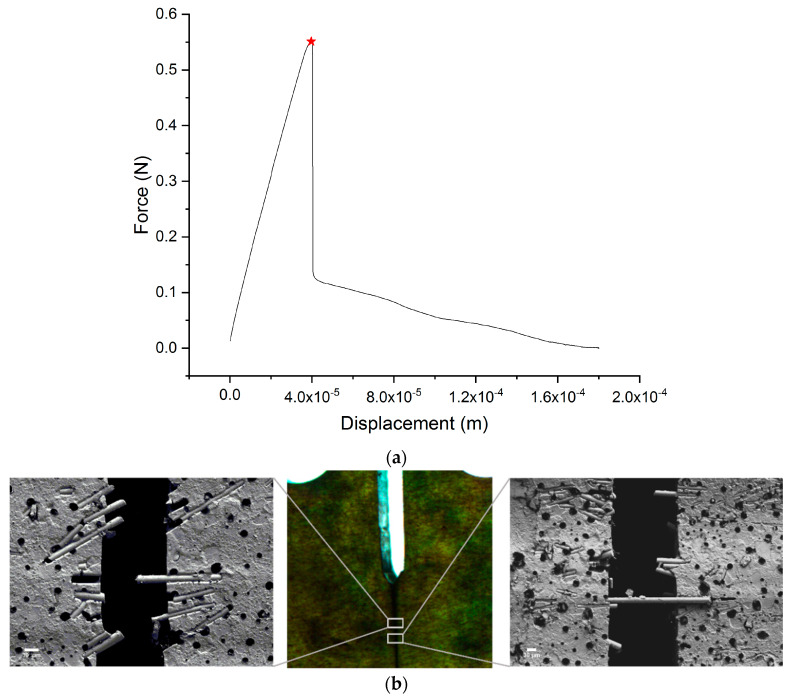
Toughening mechanisms. (**a**) Force-displacement curve of a pull-out test performed on a single magnetized glass fiber (MGF) and Epoxy (**b**) Scanning Electron Microscopy (SEM) images of the crack walls after fracture, showing some of the main toughening mechanisms involved in the extrinsic toughening of the composite: fiber pull-out and fiber bridging. Note: the black spots in the SEM images are traces of artifacts in the Silicon mold surface; they do not indicate voids in the specimen volume.

**Table 1 materials-13-02415-t001:** Four categories of specimens, defined by their curing setup and by the ratio between the concentrating field (*H_c_*) and the bias field (*H_b_*) acting on the sample during curing.

Sample	Description (see Figure 2)	Orientation (see Figure 6)	Concentration	*H_c_*/*H_b_*
Control	None	Random	Uniform	-
Setup 1	A pair of small permanent magnets	Unidirectional (Z)	Concentrated near the crack-tip	∞
Setup 2	Solenoid and rotated motor	Unidirectional (X)	Uniform	0
Setup 3	A pair of small permanent magnets + Bias magnetic field	Unidirectional (X)	Concentrated near the crack-tip	> 0

**Table 2 materials-13-02415-t002:** Results summary of fracture toughness vs. weight fraction for uniformly distributed and randomly oriented fibers. K_IC,0_ indicates the toughness of an epoxy-only specimen.

wt %	*K_IC_* [MPa·m^0.5^]	*K_IC_*/*K_IC_*_,0_
0	0.86 ± 0.21	1.00
2	0.83 ± 0.28	0.97
10	0.96 ± 0.14	1.12
20	1.20 ± 0.15	1.40
30	1.48 ± 0.16	1.72

**Table 3 materials-13-02415-t003:** Comparison of the fracture toughness of EP/MGF composite (10 wt %) under different magnetic configurations. The magnetic configuration is depicted as the ratio of the concentration field and the bias field; $\hat{\theta }$ is the average orientation angle as calculated according to Equation (3).

*H_c_*/*H_b_* [G/G]	$$\hat{\theta }[\deg.]$$	*K_IC_* [MPa·m^0.5^]	*K_IC_*/*K*_*IC*,0_
0 (solenoid)	0	1.34 ± 0.19	1.56
0.1	0.6	1.38 ± 0.04	1.60
0.3	5	1.60 ± 0.18	1.86
0.5	14	1.28 ± 0.27	1.49
1.3	59	1.17 ± 0.22	1.36
1.5	66	1.17 ± 0.18	1.36
∞ (no bias)	90	0.80 ± 0.12	0.93
control	38	0.96 ± 0.14	1.12

**Table 4 materials-13-02415-t004:** Composite Properties.

Symbol	Description	Method	Value	Units
*d_f_*	Short fiber diameter	SEM	16	μm
*l_f_*	Short fiber average length	μCT	220 ± 50	μm
*E_m_*	Matrix tensile modulus	Tensile (dog-bone)	1.22 ± 0.14	GPa
*E_f_*	Fiber tensile modulus	Tensile	80 ± 12	GPa
*σ^y^_0_*	Matrix yield stress	Tensile (dog-bone)	72.6 ± 4.1	MPa
*σ_m_^max^_0_*	Matrix ultimate stress	Tensile (dog-bone)	100.8 ± 0.14	MPa
*σ_f_*	Fiber ultimate strength	Tensile (Weibull)	3248 ± 50	MPa
*τ_i_*	Interfacial shear strength	Pull-out	34.6 ± 2.9	MPa
*G_i_*	fiber-matrix surface energy	Pull-out	1.6 ± 0.8	kJ/m^2^
*l_c_*	Critical length	Calculated [42]	740	μm
*r_p_*	Radius of plasticity	Calculated [13]	28	μm
*D_pz_*	Process-zone size	Calculated [34]	650	μm

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
