# Peer review of "Efficient Toughening of Short-Fiber Composites Using Weak Magnetic Fields"

_materials, 2020, doi:10.3390/ma13102415_

Round 1

Reviewer 1 Report

In this work, the authors use magnetic fields to orient milled glass fiber within polymeric materials to enhance toughness. Though the ideas of orienting short fibers with magnetic fields have already been presented elsewhere, this study provides a very holistic report on a specific materials system that brings insight to the field. I recommend this manuscript may be granted acceptance after the authors address the following comments.

  1. In the first sentences the authors claim discontinuous composites can be tougher than continuous. This statement should be defended, toned down, or withdrawn. In general, discontinuous GFRP are a far cry from the toughness of continuous GFRP. KIC's of 1.5 that the authors report need to be compared with around 15 of typical continuous GFRP. 
  2. The failure in Liberty Ships did not spur a re-evaluation of Griffith's seminal work, but rather reminded people of the importance of it. Liberty ships was about not understanding the brittle to ductile transition in BCC metals well enough. Seems like history shouldn't be altered unless the authors can defend it better.
  3. Figure 5 leaves a lot to be desired. The concentration of fiber is established with light intensity - but this is just qualitative. This can be made quantitative using a calibration relationships derived from a series of samples of known concentration and measured intensity. Further, this work looks at different concentration gradients, yet the details of those concentration gradients aren't established. This seems to be required to make any sort of conclusion.
  4. The micro-ct is a nice touch for this work.
  5. The authors report that surface treatment of the gf doesn't affect the interface - which raises eyebrows. This either means that the fibers connect well with the polymer as is - or that the fibers don't connect well in either case. The authors need more here. They don't report pull out tests for the sized fibers, they don't detail what the sizing is, etc...
  6. The actual calculation of the KIC needs to be better detailed and illustrated. The authors supply an equation for the pull-out test, but they go on and report KIC for different setups. More detail is needed to understand how the KIC is calculated and under what assumptions.

Reviewer 2 Report

This manuscript reports the research on improving the mechanical properties of epoxy matrix by using short glass fiber. It creatively uses weak magnetic field to change the distribution and orientation of the fiber in the epoxy matrix at the same time, makes full use of the toughening mechanism of the fiber, and greatly improves the fracture toughness of the epoxy matrix. After careful review of the manuscript, the following questions were raised:

  1. Can the adsorption degree of iron oxide particles and the magnetization degree of fibers be consistent in each material preparation process?Will this affect the repeatability of the experiment?
  2. The names of the three Setups in this manuscript do not correspond to the contents, such as table 1, figure 2 and figure 9.
  3. The sample features represented by W, a and b should be best indicated in the figure, so that readers can better understand the meaning of each parameter (figure 3).
  4. The third column in figure 6 should be the XZ plane, not the YZ plane.
  5. The table in this manuscript should be a three-line table.
  6. What is the reason for choosing a composite containing 10% wt fiber for magnetic field assisted experiment? Why not choose one with higher fracture toughness?
  7. Will the fracture toughness continue to increase as the addition of fiber continues to increase above 30%wt? What is the content of glass fiber when the fracture toughness of the composite reaches its peak?  (Figure8)
  8. What are the large number of black spheres in Figure 10?

Reviewer 3 Report

In this work, the authors introduce a toughening mechanism in short fiber reinforced composites based on the realignment and concentration of magnetic fibers during curing of the matrix to regions with high stresses. In general, the paper is very well written; it is easy to follow all the basic concepts. I would say that the paper is written with a non-expert reader in mind since it is straightforward to follow all the underlining mechanical concepts of fracture mechanics and procedures applied by the authors. Unfortunately, the enthusiasm for reviewing this paper was dampened due to the incremental nature of the work. The work of fiber realignment using magnetic fields is not new, and it has been used before not only to increase toughness in composites but also to create actuation and add multifunctionality. Following are a few key references that the authors missed:

  1. Le Ferrand, H., Bouville, F., Niebel, T.P. and Studart, A.R., 2015. Magnetically assisted slip casting of bioinspired heterogeneous composites. Nature materials, 14(11), pp.1172-1179.
  2. Bargardi, F.L., Le Ferrand, H., Libanori, R. and Studart, A.R., 2016. Bio-inspired self-shaping ceramics. Nature communications, 7(1), pp.1-8.
  3. Schmied, J.U., Le Ferrand, H., Ermanni, P., Studart, A.R. and Arrieta, A.F., 2017. Programmable snapping composites with bio-inspired architecture. Bioinspiration & biomimetics, 12(2), p.026012.
  4. Martin, J.J., Fiore, B.E. and Erb, R.M., 2015. Designing bioinspired composite reinforcement architectures via 3D magnetic printing. Nature communications, 6(1), pp.1-7.
  5. Erb, R.M., Libanori, R., Rothfuchs, N. and Studart, A.R., 2012. Composites reinforced in three dimensions by using low magnetic fields. Science, 335(6065), pp.199-204.

Some of the methods presented in these references allow localized and realigned fibers in complex architectures. The authors have to differentiate their work from these and other references before any consideration for publication. 

Also, the authors should improve the quality of the figures. For instance, Figures 1a-b are pixelated. The axes are difficult to read. The font is too small. Inset in figure 1b is not legible. Similar comments apply for the other figures in the manuscript.

Round 2

Reviewer 1 Report

Comments sufficiently addressed.

Reviewer 3 Report

The authors have addressed the concerns that this reviewer has about the novelty of the work.

The authors' argument that previous works did not demonstrate the translation of fibers to achieve locally varying filler concentration. The reviewer agrees that this angle has not been exploited previously, and the study of this will be of interest to the materials community.